# Differentiate Everything with a Reversible Embedded Domain-Specific Language

## Abstract

Reverse-mode automatic differentiation (AD) suffers from the issue of having too much space overhead to trace back intermediate computational states for back-propagation. The traditional method to trace back states is called checkpointing that stores intermediate states into a global stack and restore state through either stack pop or re-computing. The overhead of stack manipulations and re-computing makes the general purposed (not tensor-based) AD engines unable to meet many industrial needs. Instead of checkpointing, we propose to use reverse computing to trace back states by designing and implementing a reversible programming eDSL, where a program can be executed bi-directionally without implicit stack operations. The absence of implicit stack operations makes the program compatible with existing compiler features, including utilizing existing optimization passes and compiling the code as GPU kernels. We implement AD for sparse matrix operations and some machine learning applications to show that our framework has the state-of-the-art performance.

## 1 Introduction

Most of the popular automatic differentiation (AD) tools in the market, such as TensorFlow (Abadi et al., 2015), Pytorch (Paszke et al., 2017), and Flux (Innes et al., 2018) implements reverse mode AD at the tensor level to meet the need in machine learning. Later, People in the scientific computing domain also realized the power of these AD tools, they use these tools to solve scientific problems such as seismic inversion (Zhu et al., 2020), variational quantum circuits simulation (Bergholm et al., 2018; Luo et al., 2019) and variational tensor network simulation (Liao et al., 2019; Roberts et al., 2019). To meet the diverse need in these applications, one sometimes has to define backward rules manually, for example

1. To differentiate sparse matrix operations used in Hamiltonian engineering (Hao Xie & Wang), people defined backward rules for sparse matrix multiplication and dominant eigensolvers (Golub & Van Loan, 2012),

2. In tensor network algorithms to study the phase transition problem (Liao et al., 2019; Seeger et al., 2017; Wan & Zhang, 2019; Hubig, 2019), people defined backward rules for singular value decomposition (SVD) function and QR decomposition (Golub & Van Loan, 2012).

Instead of defining backward rules manually, one can also use a general purposed AD (GP-AD) framework like Tapenade (Hascoet & Pascual, 2013), OpenAD (Utke et al., 2008) and Zygote (Innes, 2018; Innes et al., 2019). Researchers have used these tools in practical applications such as bundle adjustment (Shen & Dai, 2018) and earth system simulation (Forget et al., 2015), where differentiating scalar operations is important. However, the power of these tools are often limited by their relatively poor performance. In many practical applications, a program might do billions of computations. In each computational step, the AD engine might cache some data for backpropagation. (Griewank & Walther, 2008) Frequent caching of data slows down the program significantly, while the memory usage will become a bottleneck as well. Caching implicitly also make these frameworks incompatible with kernel functions. To avoid such issues, we need a new GP-AD framework that does not cache automatically for users.

In this paper, we propose to implement the reverse mode AD on a reversible (domain-specific) programming language (Perumalla, 2013; Frank, 2017), where intermediate states can be traced

backward without accessing an implicit stack. Reversible programming allows people to utilize the reversibility to reverse a program. In machine learning, reversibility is proven to substantially decrease the memory usage in unitary recurrent neural networks (MacKay et al., 2018), normalizing flow (Dinh et al., 2014), hyper-parameter learning (Maclaurin et al., 2015) and residual neural networks (Gomez et al., 2017; Behrmann et al., 2018). Reversible programming will make these happen naturally. The power of reversible programming is not limited to handling these reversible applications, any program can be written in a reversible style. Converting an irreversible program to the reversible form would cost overheads in time and space. Reversible programming provides a flexible time-space trade-off scheme that different with checkpointing (Griewank, 1992; Griewank & Walther, 2008; Chen et al., 2016), *reverse computing* (Bennett, 1989; Levine & Sherman, 1990), to let user handle these overheads explicitly.

There have been many prototypes of reversible languages like Janus (Lutz, 1986), R (not the popular one) (Frank, 1997), Erlang (Lanese et al., 2018) and object-oriented ROOPL (Haulund, 2017). In the past, the primary motivation to study reversible programming is to support reversible computing devices (Frank & Knight Jr, 1999) such as adiabatic complementary metal-oxide-semiconductor (CMOS) (Koller & Athas, 1992), molecular mechanical computing system (Merkle et al., 2018) and superconducting system (Likharev, 1977; Semenov et al., 2003; Takeuchi et al., 2014; 2017), and these reversible computing devices are orders more energy-efficient. Landauer proves that only when a device does not erase information (i.e. reversible), its energy efficiency can go beyond the thermal dynamic limit. (Landauer, 1961; Reeb & Wolf, 2014) However, these reversible programming languages can not be used directly in real scientific computing, since most of them do not have basic elements like floating point numbers, arrays, and complex numbers. This motivates us to build a new embedded domain-specific language (eDSL) in Julia (Bezanson et al., 2012; 2017) as a new playground of GP-AD.

In this paper, we first compare the time-space trade-off in the optimal checkpointing and the optimal reverse computing in Sec. 2. Then we introduce the language design of NiLang in Sec. 3. In Sec. 4, we explain the implementation of automatic differentiation in NiLang. In Sec. 5, we benchmark the performance of NiLang's AD with other AD software and explain why it is fast.

## 2 REVERSE COMPUTING AS AN ALTERNATIVE TO CHECKPOINTING

One can use either checkpointing or reverse computing to trace back intermediate states of a $T$-step computational process $s_1 = f_1(s_0), s_2 = f_2(s_1), \ldots, s_T = f_T(s_{T-1})$ with a run-time memory $S$. In the checkpointing scheme, the program first takes snapshots of states at certain time steps $S = \{s_a, s_b, \ldots\}, 1 \leq a < b < \ldots \leq T$ by running a forward pass. When retrieving a state $s_k$, if $s_k \in S$, just return this state, otherwise, return $\max_j s_{j<k} \in S$ and re-compute $s_k$ from $s_j$. In the reverse computing scheme, one first writes the program in a reversible style. Without prior knowledge, a regular program can be transpiled to the reversible style is by doing the transformation in Listing. 1.

Listing 1: Transpiling a regular code to the reversible code without prior knowledge.

```
s₁ += f₁(s₀)
s₂ += f₂(s₁)
...
s_T += f_T(s_{T-1})
```

Listing 2: The reverse of Listing. 1

```
s_T -= f_T(s_{T-1})
...
s₂ -= f₂(s₁)
s₁ -= f₁(s₀)
```

Then one can visit states in the reversed order by running the reversed program in Listing. 2, which erases the computed results from the tail. One may argue that easing through uncomputing is not necessary here. This is not true for a general reversible program, because the intermediate states might be mutable and used in other parts of the program. It is easy to see, both checkpointing and reverse computing can trace back states without time overhead, but both suffer from a space overhead that linear to time (Table 1). The checkpointing scheme snapshots the output in every step, and the reverse computing scheme allocates extra storage for storing outputs in every step. On the other side, only checkpointing can achieve a zero space overhead by recomputing everything from the beginning $s_0$, with a time complexity $O(T^2)$. The minimum space complexity in reverse

computing is $O(S \log(T/S))$ (Bennett, 1989; Levine & Sherman, 1990; Perumalla, 2013), with time complexity $O(T^{1.585})$.

| Method | most time efficient (Time/Space) | most space efficient (Time/Space) |
|---|---|---|
| Checkpointing | $O(T)/O(T + S)$ | $O(T^2)/O(S)$ |
| Reverse computing | $O(T)/O(T + S)$ | $O(T(\frac{T}{S})^{0.585})/O(S \log(\frac{T}{S}))$ |

Table 1: $T$ and $S$ are the time and space of the original irreversible program. In the "Reverse computing" case, the reversibility of the original program is not utilized.

The difference in space overheads can be explained by the difference of the optimal checkpointing and optimal reverse computing algorithms. The optimal checkpointing algorithm that widely used in AD is the treeverse algorithm in Fig. 1(a). This algorithm partitions the computational process **binomially** into $d$ sectors. At the beginning of each sector, the program snapshots the state and push it into a global stack, hence the memory for checkpointing is $dS$. The states in the last sector are retrieved by the above space-efficient $O(T^2)$ algorithm. After that, the last snapshot can be freed and the program has one more quota in memory. With the freed memory, the second last sector can be further partition into two sectors. Likewise, the $l$th sectors is partitioned into $l$ sub-sectors, where $l$ is the sector index counting from the tail. Recursively apply this treeverse algorithm $t$ times until the sector size is 1. The approximated overhead in time and space are

$$T_c = tT, S_c = dS, \tag{1}$$

where $T = \eta(t, d)$ holds. By carefully choosing either a $t$ or $d$, the overhead in time and space can be both logarithmic.

On the other side, the optimal time-space trade-off scheme in reverse computing is the Bennett's algorithm illustrated in Fig. 1 (b). It evenly **evenly** partition the program into $k$ sectors. The program marches forward ($P$ process) for $k$ steps to obtain the final state $s_{k+1}$, then backward ($Q$ process) from the $k - 1$th step to erase the states in between $s_{1<i<k}$. This process is also called the *compute-copy-uncompute* process. Recursively apply the compute-copy-uncompute process for each sector until each $P/Q$ process contains only one unit computation. the time and space complexities are

$$T_r = T \left(\frac{T}{S}\right)^{\frac{\ln(2-(1/k))}{\ln k}}, S_r = \frac{k-1}{\ln k} S \log \frac{T}{S}. \tag{2}$$

Here, the overhead in time is polynomial, which is worse than the treeverse algorithm. The treeverse like partition does not apply here because the first sweep to create initial checkpoints without introducing any space overheads is not possible in reversible computing. The pseudo-code of Bennett's time-space trade-off algorithm is shown in Listing. 3.

Listing 3: The Bennett's time-space trade-off scheme. The first argument $\{s_1, \ldots\}$ is the collection of states, k is the number of partitions, i and len are the starting point and length of the working sector. A function call changes variables inplace. "~" is the symbol of uncomputing, which means undoing a function call. Statement $s_{i+1} \leftarrow 0$ allocates a zero state and add it to the state collection. Its inverse $s_{i+1} \rightarrow 0$ discards a zero cleared state from the collection. Its NiLang implementation is in Appendix A.

```
bennett({s₁,…}, k, i, len)
    if len == 1
        s_{i+1} ← 0
        f_i(s_{i+1}, s_i)
    else
        # P process that calls the forward program k steps
        bennett({s₁,…}, k, i+len÷k*(j-1), len÷k) for j=1,2,…,k
        # Q process that calls the backward program k-1 steps
        ~bennett({s₁,…}, k, i+len÷k*(j-1), len÷k) for j=k-1,k-2,…,1
```

The reverse computing does not show advantage from the above complexity analysis. But we argue the this analysis is from the worst case, which are very different to the practical using cases. *First,*

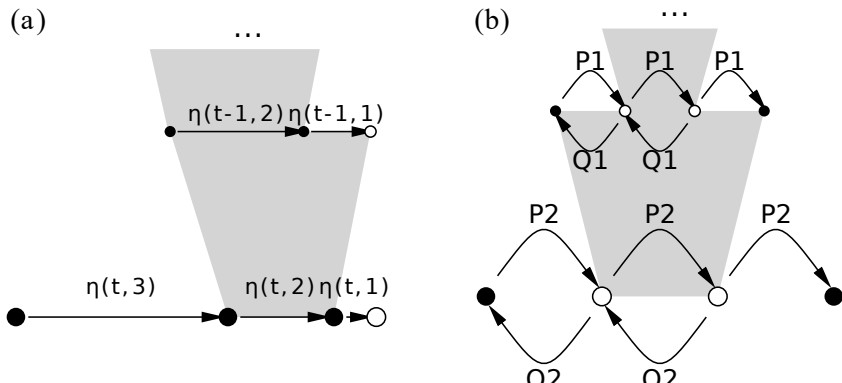

Figure 1: (a) Treeverse algorithm for optimal checkpointing. (Griewank, 1992) $\eta(\tau, \delta) \equiv \binom{\tau + \delta}{\delta} = \frac{(\tau+\delta)!}{\tau!\delta!}$ is the binomial function. (b) Bennett's time space trade-off scheme for reverse computing. (Bennett, 1973; Levine & Sherman, 1990) $P$ and $Q$ are computing and uncomputing respectively. The pseudo-code is defined in Listing. 3.

reverse computing can make use of the reversibility to save memory. In Appendix B.2, we show how to implement a unitary matrix multiplication without introducing overheads in space and time. *Second*, reverse computing does not allocate automatically for users, user can optimize the memory access patterns for their own devices like GPU. *Third*, reverse computing is compatible with effective codes, so that it fits better with modern languages. In Appendix B.1, we show how to manipulate inplace functions on arrays with NiLang. *Fourth*, reverse computing can utilize the existing compiler to optimize the code because it does not introduce global stack operations that harm the purity of functions. *Fifth*, reverse computing encourages users to think reversibly. In Appendix B.3, we show reversible thinking can lead the user to a constant memory, constant time implementation of chained multiplication algorithms.

## 3 LANGUAGE DESIGN

NiLang is an embedded domain-specific language (eDSL) NiLang built on top of the host language Julia (Bezanson et al., 2012; 2017). Julia is a popular language for scientific programming and machine learning. We choose Julia mainly for speed. Julia is a language with high abstraction, however, its clever design of type inference and just in time compiling make it has a C like speed. Meanwhile, it has rich features for meta-programming. Its package for pattern matching MLStyle allows us to define an eDSL in less than 2000 lines. Comparing with a regular reversible programming language, NiLang features array operations, rich number systems including floating-point numbers, complex numbers, fixed-point numbers, and logarithmic numbers. It also implements the compute-copy-uncompute (Bennett, 1973) macro to increase code reusability. Besides the above reversible hardware compatible features, it also has some reversible hardware incompatible features to meet the practical needs. For example, it views the floating-point + and − operations as reversible. It also allows users to extend instruction sets and sometimes inserting external statements. These features are not compatible with future reversible hardware. NiLang's compiling process, grammar and operational semantics are described in Appendix G. The source code is also available online, we will put a link here after the anonymous open review session. By the time of writing, the version of NiLang is v0.7.3.

### 3.1 REVERSIBLE FUNCTIONS AND INSTRUCTIONS

Mathematically, any irreversible mapping `y = f(args...)` can be trivially transformed to its reversible form `y += f(args...)` or `y ⊻= f(args...)` (⊻ is the bit-wise XOR), where `y` is a pre-emptied variable. But in numeric computing with finite precision, this is not always true. The reversibility of arithmetic instruction is closely related to the number system. For integer and fixed

point number system, `y += f(args...)` and `y -= f(args...)` are rigorously reversible. For logarithmic number system and tropical number system (Speyer & Sturmfels, 2009), `y *= f(args...)` and `y /= f(args...)` as reversible (not introducing the zero element). While for floating point numbers, none of the above operations are rigorously reversible. However, for convenience, we ignore the round-off errors in floating-point + and – operations and treat them on equal footing with fixed-point numbers in the following discussion. In Appendix F, we will show doing this is safe in most cases provided careful implementation. Other reversible operations includes `SWAP`, `ROT`, `NEG` et. al., and this instruction set is extensible. One can define a reversible multiplier in NiLang as in Listing. 4.

Listing 4: A reversible multiplier

```julia
julia> using NiLang

julia> @i function multiplier(y!::Real, a::Real, b::Real)
           y! += a * b
       end

julia> multiplier(2, 3, 5)
(17, 3, 5)

julia> (~multiplier)(17, 3, 5)
(2, 3, 5)
```

Macro `@i` generates two functions that are reversible to each other, `multiplier` and `~multiplier`, each defines a mapping $\mathbb{R}^3 \to \mathbb{R}^3$. The ! after a symbol is a part of the name, as a conversion to indicate the mutated variables.

## 3.2 Reversible memory management

A distinct feature of reversible memory management is that the content of a variable must be known when it is deallocated. We denote the allocation of a pre-emptied memory as $x \leftarrow 0$, and its inverse, deallocating a **zero emptied** variable, as $x \to 0$. An unknown variable can not be deallocate, but can be pushed to a stack pop out later in the uncomputing stage. If a variable is allocated and deallocated in the local scope, we call it an ancilla. Listing. 5 defines the complex valued accumulative log function.

Listing 5: Reversible complex valued log function $y += \log(|x|) + i\mathrm{Arg}(x)$.

```julia
@i @inline function (:+=)(log)(y!::Complex{T
    }, x::Complex{T}) where T
    n ← zero(T)
    n += abs(x)

    y!.re += log(n)
    y!.im += angle(x)

    n -= abs(x)
    n → zero(T)
end
```

Listing 6: Compute-copy-uncompute version of Listing 5

```julia
@i @inline function (:+=)(log)(y!::Complex{T
    }, x::Complex{T}) where T
    @routine begin
        n ← zero(T)
        n += abs(x)
    end
    y!.re += log(n)
    y!.im += angle(x)
    ~@routine
end
```

Here, the macro `@inline` tells the compiler that this function can be inlined. One can input "←" and "→" by typing "\leftarrow[TAB KEY]" and "\rightarrow[TAB KEY]" respectively in a Julia editor or REPL. NiLang does not have immutable structs, so that the real part `y!.re` and imaginary `y!.im` of a complex number can be changed directly. It is easy to verify that the bottom two lines in the function body are the inverse of the top two lines. i.e., the bottom two lines *uncomputes* the top two lines. The motivation of uncomputing is to zero clear the contents in ancilla `n` so that it can be deallocated correctly. *Compute-copy-uncompute* is a useful design pattern in reversible programming so that we created a pair of macros `@routine` and `~@routine` for it. One can rewrite the above function as in Listing. 6.

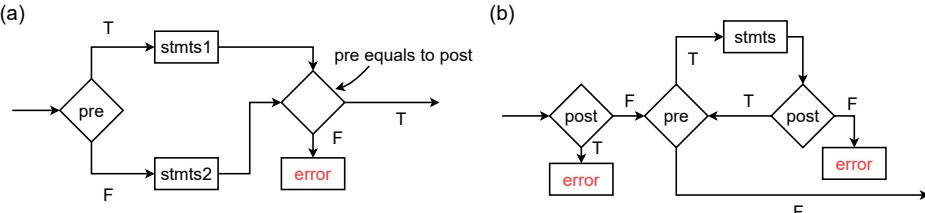

Figure 2: The flow chart for reversible (a) `if` statement and (b) `while` statement. "pre" and "post" represents precondition and postcondition respectively. The assersion errors are thrown to the host language instead of handling them in NiLang.

## 3.3 REVERSIBLE CONTROL FLOWS

One can define reversible `if`, `for` and `while` statements in a reversible program. Fig. 2 (a) shows the flow chart of executing the reversible `if` statement. There are two condition expressions in this chart, a precondition and a postcondition. The precondition decides which branch to enter in the forward execution, while the postcondition decides which branch to enter in the backward execution. The pseudo-code for the forward and backward passes are shown in Listing. 7 and Listing. 8.

Listing 7: Translating a reversible `if` statement (forward)

```
branchkeeper = precondition
if precondition
    branch A
else
    branch B
end
assert branchkeeper ==  postcondition
```

Listing 8: Translating a reversible `if` statement (backward)

```
branchkeeper = postcondition
if postcondition
    ~(branch A)
else
    ~(branch B)
end
assert branchkeeper ==  precondition
```

Fig. 2 (b) shows the flow chart of the reversible `while` statement. It also has two condition expressions. Before executing the condition expressions, the program presumes the postcondition is false. After each iteration, the program asserts the postcondition to be true. To reverse this statement, one can exchange the precondition and postcondition, and reverse the body statements. The pseudo-code for the forward and backward passes are shown in Listing. 9 and Listing. 10.

Listing 9: Translating a reversible `while` statement (forward)

```
assert postcondition == false
while precondition
    loop body
    assert postcondition == true
end
```

Listing 10: Translating a reversible `while` statement (backward)

```
assert precondition == false
while postcondition
    ~(loop body)
    assert precondition == true
end
```

The reversible `for` statement is similar to the irreversible one except that after execution, the program will assert the iterator to be unchanged. To reverse this statement, one can exchange `start` and `stop` and inverse the sign of `step`. Listing. 11 computes the Fibonacci number recursively and reversibly.

Listing 11: Computing Fibonacci number recursively and reversibly.

```
@i function rrfib(out!, n)
    @invcheckoff if (n >= 1, ~)
        counter ← 0
        counter += n
        while (counter > 1, counter!=n)
            rrfib(out!, counter-1)
            counter -= 2
        end
        counter -= n % 2
        counter → 0
    end
    out! += 1
end
```

Here, `out!` is an integer initialized to `0` for storing outputs. The precondition and postcondition are wrapped into a tuple. In the `if` statement, the postcondition is the same as the precondition, hence we omit the postcondition by inserting a "~" in the second field for "copying the precondition in this field as the postcondition". In the while statement, the postcondition is true only for the initial loop. Once code is proven correct, one can turn off the reversibility check by adding `@invcheckoff` before a statement. This will remove the reversibility check and make the code faster and compatible with GPU kernels (kernel functions can not handle exceptions).

## 4 REVERSIBLE AUTOMATIC DIFFERENTIATION

### 4.1 BACK PROPAGATION

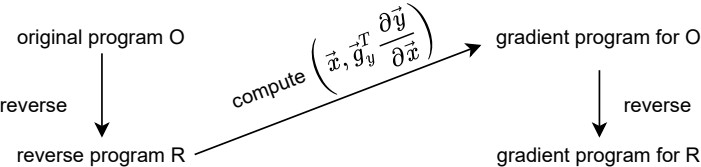

We decompose the problem of reverse mode AD into two sub-problems, **reversing the code** and **computing** $\frac{\partial[\text{single input}]}{\partial[\text{multiple outputs}]}$. Reversing the code is trivial in reversible programming. Computing the gradient here is similar to forward mode automatic differentiation that computes $\frac{\partial[\text{multiple outputs}]}{\partial[\text{single input}]}$. Inspired by the Julia package ForwardDiff (Revels et al., 2016), we use the operator overloading technique to differentiate the program efficiently. In the backward pass, we wrap each output variable with a composite type `GVar` that containing an extra gradient field, and feed it into the reversed generic program. Instructions are multiple dispatched to corresponding gradient instructions that update the gradient field of `GVar` at the meantime of uncomputing. By reversing this gradient program, we can obtain the gradient program for the reversed program too. One can define the adjoint ("adjoint" here means the program for back-propagating gradients) of a primitive instruction as a reversible function on **either** the function itself or its reverse since the adjoint of a function's reverse is equivalent to the reverse of the function's adjoint.

$$f:(\vec{x}, \vec{g}_x) \rightarrow (\vec{y}, \vec{g}_x^T \frac{\partial \vec{x}}{\partial \vec{y}}) \tag{3}$$

$$f^{-1}:(\vec{y}, \vec{g}_y) \rightarrow (\vec{x}, \vec{g}_y^T \frac{\partial \vec{y}}{\partial \vec{x}}) \tag{4}$$

It can be easily verified by applying the above two mappings consecutively, which turns out to be an identity mapping considering $\frac{\partial \vec{y}}{\partial \vec{x}} \frac{\partial \vec{x}}{\partial \vec{y}} = \mathbb{1}$.

The implementation details are described in Appendix C. In most languages, operator overloading is accompanied with significant overheads of function calls and object allocation and deallocation. But in a language with type inference and just in time compiling like Julia, the boundary between two approaches are vague. The compiler inlines small functions, packs an array of constant sized immutable objects into a continuous memory, and truncates unnecessary branches automatically.

## 4.2 HESSIANS

Combining forward mode AD and reverse mode AD is a simple yet efficient way to obtain Hessians. By wrapping the elementary type with `Dual` defined in package ForwardDiff and throwing it into the gradient program defined in NiLang, one obtains one row/column of the Hessian matrix. We will use this approach to compute Hessians in the graph embedding benchmark in Sec. D.2.

## 4.3 CUDA KERNELS

CUDA programming is playing a significant role in high-performance computing. In Julia, one can write GPU compatible functions in native Julia language with KernelAbstractions. (Besard et al., 2017) Since NiLang does not push variables into stack automatically for users, it is safe to write differentiable GPU kernels with NiLang. We will differentiate CUDA kernels with no more than extra 10 lines in the bundle adjustment benchmark in Sec. 5.

## 5 BENCHMARKS

We benchmark our framework with the state-of-the-art GP-AD frameworks, including source code transformation based Tapenade and Zygote and operator overloading based ForwardDiff and ReverseDiff. Since most tensor based AD software like famous TensorFlow and PyTorch are not designed for the using cases used in our benchmarks, we do not include those package to avoid an unfair comparison. In the following benchmarks, the CPU device is Intel(R) Xeon(R) Gold 6230 CPU @ 2.10GHz, and the GPU device is NVIDIA Titan V. For NiLang benchmarks, we have turned the reversibility check off to achieve a better performance.

We reproduced the benchmarks for Gaussian mixture model (GMM) and bundle adjustment in Srajer et al. (2018) by re-writing the programs in a reversible style. We show the results in Fig. 3. The Tapenade data is obtained by executing the docker file provided by the original benchmark, which provides a baseline for comparison.

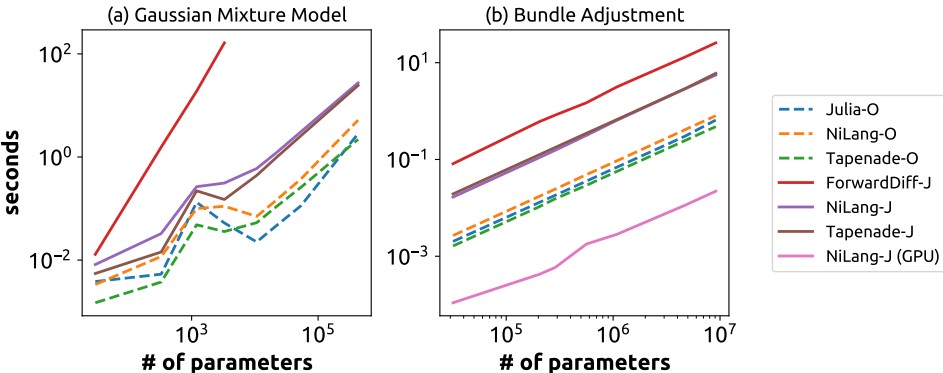

Figure 3: Absolute runtimes in seconds for computing the objective (-O) and Jacobians (-J). (a) GMM with 10k data points, the loss function has a single output, hence computing Jacobian is the same as computing gradient. ForwardDiff data is missing due to not finishing in limited time. The NiLang GPU data is missing because we do not write kernel here. (b) Bundle adjustment.

NiLang's objective function is ~2× slower than normal code due to the uncomputing overhead. In this case, NiLang does not show advantage to Tapenade in obtaining gradients, the ratio between computing the gradients and the objective function are close. This is because the bottleneck of this

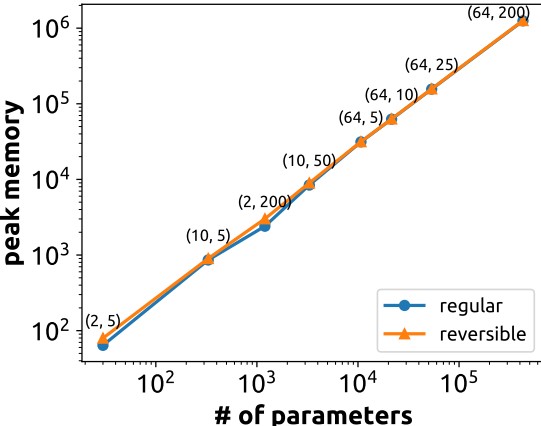

Figure 4: Peak memory of running the original and the reversible GMM program. The labels are $(d, k)$ pairs.

model is the matrix vector multiplication, traditional AD can already handle this function well. The extra memory used to reverse the program is negligible comparing to the original program as shown in Fig. 4. The backward pass is not shown here, it is just two times the reversible program in order to store gradients. The data is obtained by counting the main memory allocations in the program manually. The analytical expression of memory usage in unit of floating point number is

$$S = (2 + d^2)k + 2d + P, \tag{5}$$

$$S_r = (3 + d^2 + d)k + 2\log_2 k + P, \tag{6}$$

where $d$ and $k$ are the size and number of covariance matrices. $P = \frac{d(d+1)}{2}k + k + dk$ is the size of parameter space. The memory of the dataset $(d \times N)$ is not included because it will scale as $N$. Due to the hardness of estimating peak memory usage, the Tapenade data is missing here. The ForwardDiff memory usage is approximately the original size times the batch size, where the batch size is 12 by default.

In the bundle adjustment benchmark, NiLang performs the best on CPU. We also compiled our adjoint program to GPU with no more than 10 lines of code with KernelAbstractions, which provides another ~200× speed up. Parallelizing the adjoint code requires the forward code not reading the same variable simultaneously in different threads, and this requirement is satisfied here. The peak memory of the original program and the reversible program are both equal to the size of parameter space because all "allocation"s happen on registers in this application.

One can find more benchmarks in Appendix D, including differentiating sparse matrix dot product and obtaining Hessians in the graph embedding application.

## 6 Discussion

In this work, we demonstrate a new approach to back propagates a program called reverse computing AD by designing a reversible eDSL NiLang. NiLang is a powerful tool to differentiate code from the source code level so that can be directly useful to machine learning researches. It can generate efficient backward rules, which is exemplified in Appendix E. It can also be used to differentiate reversible neural networks like normalizing flows (Kobyzev et al., 2019) to save memory, e.g. back-propagating NICE network (Dinh et al., 2014) with only constant space overheads. NiLang is most useful in solving large scale scientific problems memory efficiently. In Liu et al. (2020), people solve the ground state problem of a $28 \times 28$ square lattice spin-glass by re-writing the quantum simulator with NiLang. There are some challenges in reverse computing AD too.

- The native BLAS and convolution operations in NiLang are not optimized for the memory layout, and are too slow comparing with state of the art machine learning libraries. We

need a better implementation of these functions in the reversible programming context so that it can be more useful in training traditional deep neural networks.

- Although we show some examples of training neural networks on GPU, the shared-reading of a variable is not allowed.

- NiLang's IR does not have variable analysis. The uncomputing pass is not always necessary for the irreversible host language to deallocate memory. In many cases, the host language's variable analysis can figure this out, but it is not guarantted.

Another interesting issue is how to make use of reversible computing devices to save energy in machine learning. Reversible computing is not always more energy efficient than irreversible computing. In the time-space trade-off scheme in Sec. 2, we show the time to uncompute a unit of memory is exponential to $n$ as $Q_n = (2k - 1)^n$, and the computing energy also increases exponentially. On the other side, the amount of energy to erase a unit of memory is a constant. When $(2k - 1)^n > 1/\xi$, erasing the memory irreversibly is more energy-efficient, where $\xi$ is the energy ratio between a reversible operation (an instruction or a gate) and its irreversible counterpart.

### ACKNOWLEDGMENTS

The authors are grateful to the people who help improve this work and fundings that sponsored the research. To meet the anonymous criteria, we will add the acknowledgments after the open review session.

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

## A  NiLang implementation of Bennett's time-space trade-off algorithm

Listing 12: NiLang implementation of the Bennett's time-space trade-off scheme.

```julia
using NiLang, Test

PROG_COUNTER = Ref(0)    # (2k-1)^n
PEAK_MEM = Ref(0)        # n*(k-1)+2

@i function bennett(f::AbstractVector, state::Dict{Int,T}, k::Int, base, len) where T
    if (len == 1, ~)
        state[base+1] ← zero(T)
        f[base](state[base+1], state[base])
        @safe PROG_COUNTER[] += 1
        @safe (length(state) > PEAK_MEM[] && (PEAK_MEM[] = length(state)))
    else
        n ← 0
        n += len÷k
        # the P process
        for j=1:k
            bennett(f, state, k, base+n*(j-1), n)
        end
        # the Q process
        for j=k-1:-1:1
            ~bennett(f, state, k, base+n*(j-1), n)
        end
        n -= len÷k
        n → 0
    end
end

k = 4
n = 4
N = k ^ n
state = Dict(1=>1.0)
f(x) = x * 2.0
instructions = fill(PlusEq(f), N)

# run the program
@instr bennett(instructions, state, k, 1, N)

@test state[N+1] ≈ 2.0^N && length(state) == 2
@test PEAK_MEM[] == n*(k-1) + 2
@test PROG_COUNTER[] == (2*k-1)^n
```

The input `f` is a vector of functions and `state` is a dictionary. We also added some irreversible external statements (those marked with `@safe`) to help analyse to program.

## B  Cases where reverse computing shows advantage

### B.1  Handling effective codes

Reverse computing can handling effective codes with mutable structures and arrays. For example, the affine transformation can be implemented without any overhead.

Listing 13: Inplace affine transformation.

```
@i function i_affine!(y!::AbstractVector{T}, W::AbstractMatrix{T}, b::AbstractVector{T}, x:
    :AbstractVector{T}) where T
    @safe @assert size(W) == (length(y!), length(x)) && length(b) == length(y!)
    @invcheckoff for j=1:size(W, 2)
        for i=1:size(W, 1)
            @inbounds y![i] += W[i,j]*x[j]
        end
    end
    @invcheckoff for i=1:size(W, 1)
        @inbounds y![i] += b[i]
    end
end
```

Here, the expression following the `@safe` macro is an external irreversible statement.

## B.2 UTILIZING REVERSIBILITY

Reverse computing can utilize reversibility to trace back states without extra memory cost. For example, we can define the unitary matrix multiplication that can be used in a type of memory-efficient recurrent neural network. (Jing et al., 2016)

Listing 14: Two level decomposition of a unitary matrix.

```
@i function i_umm!(x!::AbstractArray, θ)
    M ← size(x!, 1)
    N ← size(x!, 2)
    k ← 0
    @safe @assert length(θ) == M*(M-1)/2
    for l = 1:N
        for j=1:M
            for i=M-1:-1:j
                INC(k)
                ROT(x![i,l], x![i+1,l], θ[k])
            end
        end
    end
    k → length(θ)
end
```

## B.3 ENCOURAGES REVERSIBLE THINKING

Last but not least, reversible programming encourages users to code in a memory friendly style. Since allocations in reversible programming are explicit, programmers have the flexibility to control how to allocate memory and which number system to use. For example, to compute the power of a positive fixed-point number and an integer, one can easily write irreversible code as in Listing. 15

Listing 15: A regular power function.

```
function mypower(x::T, n::Int) where T
    y = one(T)
    for i=1:n
        y *= x
    end
    return y
end
```

Listing 16: A reversible power function.

```
@i function mypower(out,x::T,n::Int) where T
    if (x != 0, ~)
        @routine begin
            ly ← one(ULogarithmic{T})
            lx ← one(ULogarithmic{T})
            lx *= convert(x)
            for i=1:n
                ly *= x
            end
        end
        out += convert(ly)
        ~@routine
    end
end
```

Since the fixed-point number is not reversible under *=, naive checkpointing would require stack operations inside a loop. With reversible thinking, we can convert the fixed-point number to logarithmic numbers to utilize the reversibility of *= as shown in Listing. 16. Here, the algorithm to convert a regular fixed-point number to a logarithmic number can be efficient. (Turner, 2010)

## C   IMPLEMENTATION OF AD IN NILANG

To backpropagate the program, we first reverse the code through source code transformation and then insert the gradient code through operator overloading. If we inline all the functions in Listing. 6, the function body would be like Listing. 17. The automatically generated inverse program (i.e. $(y, x) \rightarrow (y - \log(x), x)$) would be like Listing. 18.

Listing 17: The inlined function body of Listing. 6.

```
@routine begin
    nsq ← zero(T)
    n ← zero(T)
    nsq += x[i].re ^ 2
    nsq += x[i].im ^ 2
    n += sqrt(nsq)
end
y![i].re += log(n)
y![i].im += atan(x[i].im, x[i].re)
~@routine
```

Listing 18: The inverse of Listing. 17.

```
@routine begin
    nsq ← zero(T)
    n ← zero(T)
    nsq += x[i].re ^ 2
    nsq += x[i].im ^ 2
    n += sqrt(nsq)
end
y![i].re -= log(n)
y![i].im -= atan(x[i].im, x[i].re)
~@routine
```

To compute the adjoint of the computational process in Listing. 17, one simply insert the gradient code into its inverse in Listing. 18. The resulting inlined code is show in Listing. 19.

Listing 19: Insert the gradient code into Listing. 18, the original computational processes are highlighted in yellow background.

```
@routine begin
    nsq ← zero(GVar{T,T})
    n ← zero(GVar{T,T})

    gsqa ← zero(T)
    gsqa += x[i].re.x * 2
    x[i].re.g -= gsqa * nsq.g
    gsqa -= nsq.x * 2
    gsqa -= x[i].re.x * 2
    gsqa → zero(T)
    nsq.x += x[i].re.x ^2

    gsqb ← zero(T)
    gsqb += x[i].im.x * 2
    x[i].im.g -= gsqb * nsq.g
    gsqb -= x[i].im.x * 2
    gsqb → zero(T)
    nsq.x += x[i].im.x ^2

    @zeros T ra rb
    rta += sqrt(nsq.x)
    rb += 2 * ra
    nsq.g -= n.g / rb
    rb -= 2 * ra
    ra -= sqrt(nsq.x)
    ~@zeros T ra rb
    n.x += sqrt(nsq.x)
end

y![i].re.x -= log(n.x)
n.g += y![i].re.g / n.x

y![i].im.x-=atan(x[i].im.x,x[i].re.x)
@zeros T xy2 jac_x jac_y
xy2 += abs2(x[i].re.x)
xy2 += abs2(x[i].im.x)
jac_y += x[i].re.x / xy2
jac_x += (-x[i].im.x) / xy2
x[i].im.g += y![i].im.g * jac_y
x[i].re.g += y![i].im.g * jac_x
jac_x -= (-x[i].im.x) / xy2
jac_y -= x[i].re.x / xy2
xy2 -= abs2(x[i].im.x)
xy2 -= abs2(x[i].re.x)
~@zeros T xy2 jac_x jac_y

~@routine
```

Here, @zeros TYPE var1 var2... is the macro to allocate multiple variables of the same type. Its inverse operations starts with ~@zeros deallocates zero emptied variables. In practice, "inserting gradients" is not achieved by source code transformation, but by operator overloading. We change the element type to a composite type GVar with two fields, value x and gradient g. With multiple dispatching primitive instructions on this new type, values and gradients can be updated simultaneously. Although the code looks much longer, the computing time (with reversibility check closed) is not.

Listing 20: Time and allocation to differentiate complex valued log.

```julia
julia> using NiLang, NiLang.AD, BenchmarkTools

julia> @inline function (ir_log)(x::Complex{T}) where T
           log(abs(x)) + im*angle(x)
       end

julia> @btime ir_log(x) setup=(x=1.0+1.2im); # native code
  30.097 ns (0 allocations: 0 bytes)

julia> @btime (@instr y += log(x)) setup=(x=1.0+1.2im; y=0.0+0.0im); # reversible code
  17.542 ns (0 allocations: 0 bytes)

julia> @btime (@instr ~(y += log(x))) setup=(x=GVar(1.0+1.2im, 0.0+0.0im); y=GVar(0.1+0.2im
    , 1.0+0.0im)); # adjoint code
  25.932 ns (0 allocations: 0 bytes)
```

The performance is unreasonably good because the generated Julia code is further compiled to LLVM so that it can enjoy existing optimization passes. For example, the optimization passes can find out that for an irreversible device, uncomputing local variables n and nsq to zeros does not affect return values, so that it will ignore the uncomputing process automatically. Unlike checkpointing based approaches that focus a lot in the optimization of data caching on a global stack, NiLang does not have any optimization pass in itself. Instead, it throws itself to existing optimization passes in Julia. Without accessing the global stack, NiLang's code is quite friendly to optimization passes. In this case, we also see the boundary between source code transformation and operator overloading can be vague in a Julia, in that the generated code can be very different from how it looks.

The joint functions for primitive instructions (:+=)(sqrt) and (:-=)(sqrt) used above can be defined as in Listing. 21.

Listing 21: Adjoints for primitives (:+=)(sqrt) and (:-=)(sqrt).

```julia
@i @inline function (:-=)(sqrt)(out!::GVar, x::GVar{T}) where T
    @routine @invcheckoff begin
        @zeros T a b
        a += sqrt(x.x)
        b += 2 * a
    end
    out!.x -= a
    x.g += out!.g / b
    ~@routine
end
```

# D  MORE BENCHMARKS

## D.1  SPARSE MATRICES

We compare the call, uncall and backward propagation time used for sparse matrix dot product and matrix multiplication in Table 2. Their reversible implementations are shown in Listing. 22 and Listing. 23. The computing time for backward propagation is approximately 1.5-3 times the Julia's native forward pass, which is close to the theoretical optimal performance.

Listing 22: Reversible sparse matrix multiplication.

```julia
using SparseArrays

@i function i_dot(r::T, A::SparseMatrixCSC{T},B::SparseMatrixCSC{T}) where {T}
    m ← size(A, 1)
    n ← size(A, 2)
    @invcheckoff branch_keeper ← zeros(Bool, 2*m)
    @safe size(B) == (m,n) || throw(DimensionMismatch("matrices must have the same
      dimensions"))
    @invcheckoff @inbounds for j = 1:n
        ia1 ← A.colptr[j]
        ib1 ← B.colptr[j]
        ia2 ← A.colptr[j+1]
        ib2 ← B.colptr[j+1]
        ia ← ia1
        ib ← ib1
        @inbounds for i=1:ia2-ia1+ib2-ib1-1
            ra ← A.rowval[ia]
            rb ← B.rowval[ib]
            if (ra == rb, ~)
                r += A.nzval[ia]' * B.nzval[ib]
            end
            ## b move -> true, a move -> false
            branch_keeper[i] ⊻= ia == ia2-1 || (ib != ib2-1 && ra > rb)
            ra → A.rowval[ia]
            rb → B.rowval[ib]
            if (branch_keeper[i], ~)
                INC(ib)
            else
                INC(ia)
            end
        end
        ~@inbounds for i=1:ia2-ia1+ib2-ib1-1
            ## b move -> true, a move -> false
            branch_keeper[i] ⊻= ia == ia2-1 || (ib != ib2-1 && A.rowval[ia] > B.rowval[ib])
            if (branch_keeper[i], ~)
                INC(ib)
            else
                INC(ia)
            end
        end
    end
    @invcheckoff branch_keeper → zeros(Bool, 2*m)
end
```

Listing 23: Reversible sparse matrix dot-product.

```julia
@i function i_mul!(C::StridedVecOrMat, A::AbstractSparseMatrix, B::StridedVector{T}, α::
    Number, β::Number) where T
    @safe size(A, 2) == size(B, 1) || throw(DimensionMismatch())
    @safe size(A, 1) == size(C, 1) || throw(DimensionMismatch())
    @safe size(B, 2) == size(C, 2) || throw(DimensionMismatch())
    nzv ← nonzeros(A)
    rv ← rowvals(A)
    if (β != 1, ~)
        @safe error("only β = 1 is supported, got β = $(β).")
    end
    # Here, we close the reversibility check inside the loop to increase performance
    @invcheckoff for k = 1:size(C, 2)
        @inbounds for col = 1:size(A, 2)
            αxj ← zero(T)
            αxj += B[col,k] * α
            for j = SparseArrays.getcolptr(A)[col]:(SparseArrays.getcolptr(A)[col + 1] - 1)
                C[rv[j], k] += nzv[j]*αxj
            end
            αxj -= B[col,k] * α
        end
    end
end
```

|          | `dot`      | `mul!` (complex valued) |
|----------|------------|-------------------------|
| Julia-O  | 3.493e-04  | 8.005e-05               |
| NiLang-O | 4.675e-04  | 9.332e-05               |
| NiLang-B | 5.821e-04  | 2.214e-04               |

Table 2: Absolute runtimes in seconds for computing the objectives (O) and the backward pass (B) of sparse matrix operations. The matrix size is $1000 \times 1000$, and the element density is 0.05. The total time for computing gradients can be estimated by summing "O" and "B".

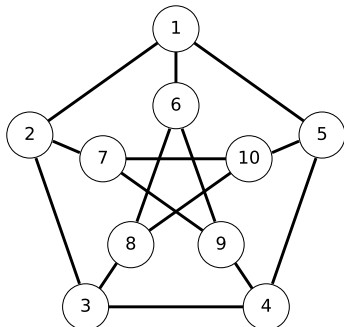

Figure 5: The Petersen graph has 10 vertices and 15 edges. We want to find a minimum embedding dimension for it.

### D.2 GRAPH EMBEDDING PROBLEM

Graph embedding can be used to find a proper representation for an order parameter (Takahashi & Sandvik, 2020) in condensed matter physics. People want to find a minimum Euclidean space dimension $k$ that a Petersen graph can embed into, that the distances between pairs of connected vertices are $l_1$, and the distance between pairs of disconnected vertices are $l_2$, where $l_2 > l_1$. The Petersen graph is shown in Fig. 5. Let us denote the set of connected and disconnected vertex pairs as $L_1$ and $L_2$, respectively. This problem can be variationally solved with the following loss.

$$
\begin{aligned}
\mathcal{L} = &\ \mathrm{Var}(\mathrm{dist}(L_1)) + \mathrm{Var}(\mathrm{dist}(L_2)) \\
&+ \exp(\mathrm{relu}(\overline{\mathrm{dist}(L_1)} - \overline{\mathrm{dist}(L_2)} + 0.1))) - 1
\end{aligned}
\tag{7}
$$

The first line is a summation of distance variances in two sets of vertex pairs, where $\mathrm{Var}(X)$ is the variance of samples in $X$. The second line is used to guarantee $l_2 > l_1$, where $\overline{X}$ means taking the average of samples in $X$. Its reversible implementation could be found in our benchmark repository.

We repeat the training for dimension $k$ from 1 to 10. In each training, we fix two of the vertices and optimize the positions of the rest. Otherwise, the program will find the trivial solution with overlapped vertices. For $k < 5$, the loss is always much higher than 0, while for $k \geq 5$, we can get a loss close to machine precision with high probability. From the $k = 5$ solution, it is easy to see $l_2/l_1 = \sqrt{2}$. An Adam optimizer with a learning rate 0.01 (Kingma & Ba) requires ~2000 steps training. The trust region Newton's method converges much faster, which requires ~20 computations of Hessians to reach convergence. Although training time is comparable, the converged precision of the later is much better.

Since one can combine ForwardDiff and NiLang to obtain Hessians, it is interesting to see how much performance we can get in differentiating the graph embedding program.

In Table 3, we show the the performance of different implementations by varying the dimension $k$. The number of parameters is $10k$. As the baseline, (a) shows the time for computing the function call. We have reversible and irreversible implementations, where the reversible program is slower

| $k$ | 2 | 4 | 6 | 8 | 10 |
|---|---|---|---|---|---|
| Julia-O | 4.477e-06 | 4.729e-06 | 4.959e-06 | 5.196e-06 | 5.567e-06 |
| NiLang-O | 7.173e-06 | 7.783e-06 | 8.558e-06 | 9.212e-06 | 1.002e-05 |
| NiLang-U | 7.453e-06 | 7.839e-06 | 8.464e-06 | 9.298e-06 | 1.054e-05 |
| NiLang-G | 1.509e-05 | 1.690e-05 | 1.872e-05 | 2.076e-05 | 2.266e-05 |
| ReverseDiff-G | 2.823e-05 | 4.582e-05 | 6.045e-05 | 7.651e-05 | 9.666e-05 |
| ForwardDiff-G | 1.518e-05 | 4.053e-05 | 6.732e-05 | 1.184e-04 | 1.701e-04 |
| Zygote-G | 5.315e-04 | 5.570e-04 | 5.811e-04 | 6.096e-04 | 6.396e-04 |
| (NiLang+F)-H | 4.528e-04 | 1.025e-03 | 1.740e-03 | 2.577e-03 | 3.558e-03 |
| ForwardDiff-H | 2.378e-04 | 2.380e-03 | 6.903e-03 | 1.967e-02 | 3.978e-02 |
| (ReverseDiff+F)-H | 1.966e-03 | 6.058e-03 | 1.225e-02 | 2.035e-02 | 3.140e-02 |

Table 3: Absolute times in seconds for computing the objectives (O), uncall objective (U), gradients (G) and Hessians (H) of the graph embedding program. $k$ is the embedding dimension, the number of parameters is $10k$.

than the irreversible native Julia program by a factor of ~2 due to the uncomputing overhead. The reversible program shows the advantage of obtaining gradients when the dimension $k \geq 3$. The larger the number of inputs, the more advantage it shows due to the overhead proportional to input size in forward mode AD. The same reason applies to computing Hessians, where the combo of NiLang and ForwardDiff gives the best performance for $k \geq 3$.

## E    PORTING NiLANG TO ZYGOTE

Zygote is a popular machine learning package in Julia. We can port NiLang's automatically generated backward rules to Zygote to accelerate some performance-critical functions. The following example shows how to speed up the backward propagation of `norm` by ~50 times.

Listing 24: Porting NiLang to Zygote.

```julia
julia> using Zygote, NiLang, NiLang.AD, BenchmarkTools, LinearAlgebra

julia> x = randn(1000);

julia> @benchmark norm'(x)
BenchmarkTools.Trial:
  memory estimate:  339.02 KiB
  allocs estimate:  8083
  --------------
  minimum time:     228.967 μs (0.00% GC)
  median time:      237.579 μs (0.00% GC)
  mean time:        277.602 μs (12.06% GC)
  maximum time:     5.552 ms (94.00% GC)
  --------------
  samples:          10000
  evals/sample:     1

julia> @i function r_norm(out::T, out2::T, x::AbstractArray{T}) where T
           for i=1:length(x)
               @inbounds out2 += x[i]^2
           end
           out += sqrt(out2)
       end

julia> Zygote.@adjoint function norm(x::AbstractArray{T}) where T
           # compute the forward with regular norm (might be faster)
           out = norm(x)
           # compute the backward with NiLang's norm, element type is GVar
           out, δy -> (grad((~r_norm)(GVar(out, δy), GVar(out^2), GVar(x))[3]),)
       end

julia> @benchmark norm'(x)
BenchmarkTools.Trial:
  memory estimate:  23.69 KiB
  allocs estimate:  2
  --------------
  minimum time:     4.015 μs (0.00% GC)
  median time:      5.171 μs (0.00% GC)
  mean time:        6.872 μs (13.00% GC)
  maximum time:     380.953 μs (93.90% GC)
  --------------
  samples:          10000
  evals/sample:     7
```

We first import the `norm` function from Julia standard library LinearAlgebra. Zygote's builtin AD engine will generate a slow code and memory allocation of 339KB. Then we write a reversible norm function `r_norm` with NiLang and port the backward function to Zygote by specifying the backward rule (the function marked with macro `Zygote.@adjoint`). Except for the speed up in computing time, the memory allocation also decreases to 23KB, which is equal to the sum of the original $x$ and the array used in backpropagation.

$$(1000 \times 8 + 1000 \times 8 \times 2)/1024 \approx 23$$

The later one has a doubled size because `GVar` has an extra gradient field.

## F    A BENCHMARK OF ROUND-OFF ERROR IN LEAPFROG

Running reversible programming with the floating pointing number system can introduce round-off errors and make the program not reversible. The quantify the effects, we use the leapfrog integrator to compute the orbitals of planets in our solar system as a benchmark. The leapfrog interations can be represented as

$$\vec{a}_i = G \frac{m_j(\vec{x}_j - \vec{x}_i)}{\|\vec{x}_i - \vec{x}_j\|^3} \tag{8}$$

$$\vec{v}_{i+1/2} = \vec{v}_{i-1/2} + \vec{a}_i \Delta t \tag{9}$$

$$\vec{x}_{i+1} = \vec{x}_i + \vec{v}_{i+1/2} \Delta t \tag{10}$$

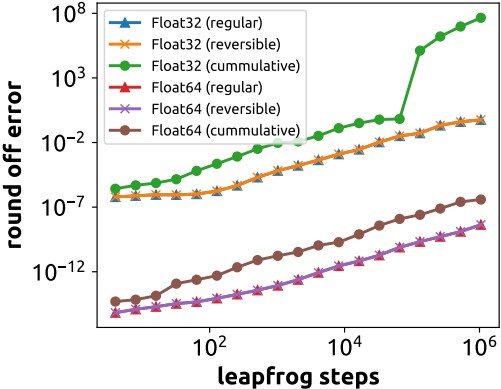

Figure 6: Round-off errors in the final axes of planets as a function of the number of time steps. "(regular)" means an irreversible program, "(reversible)" means a reversible program (Listing. 25), and "(ccumulative)" means a reversible program with the acceleration computed with ccumulative errors (Listing. 26).

where $G$ is the gravitational constant and $m_j$ is the mass of $j$th planet, $\vec{x}$, $\vec{v}$ and $\vec{a}$ are location, velocity and acceleration respectively. The first value of velocity is $v_{1/2} = a_0 \Delta t/2$. Since the dynamics of our solar system are symplectic and the leapfrog integrator is time-reversible, the reversible program does not have overheads and the evolution time can go arbitrarily long with constant memory. We compare the mean error in the final axes of the planets and show the results in Fig. 6. Errors are computed by comparing with the results computed with high precision floating-point numbers. One of the key steps that introduce round-off error is the computation of `acceleration`. If it is implemented as in Listing. 25, the round-off error does not bring additional effect in the reversible context, hence we see overlapping lines "(regular)" and "(reversible)" in the figure. This is because, when returning a dirty (not exactly zero cleared due to the floating-point round-off error) ancilla to the ancilla pool, the small remaining value will be zero-cleared automatically in NiLang. The `acceleration` function can also be implemented as in Listing. 26, where the same variable `rb` is repeatedly used for compute and uncompute, the error will accumulate on this variable. In both `Float64` (double precision floating point) and `Float32` (single precision floating point) benchmarks, the results show a much lower precision. Hence, simulating reversible programming with floating-point numbers does not necessarily make the results less reliable if one can avoid cumulative errors in the implementation.

Listing 25: Compute the acceleration. Compute and uncompute on ancilla `rc`

```
@i function :(+=)(acceleration)(y!::V3{T},
    ra::V3{T}, rb::V3{T}, mb::Real, G)
    where T
    @routine @invcheckoff begin
        @zeros T d anc1 anc2 anc3 anc4
        rc ← zero(V3{T})
        d += sqdistance(ra, rb)
        anc1 += sqrt(d)
        anc2 += anc1 ^ 3
        anc3 += G * mb
        anc4 += anc3 / anc2
        rc += rb - ra
    end
    y! += anc4 * rc
    ~@routine
end
```

Listing 26: Compute the acceleration. Compute and uncompute on the input variable `rb`.

```
@i function :(+=)(acceleration)(y!::V3{T},
    ra::V3{T}, rb::V3{T}, mb::Real, G)
    where T
    @routine @invcheckoff begin
        @zeros T d anc1 anc2 anc3 anc4
        d += sqdistance(ra, rb)
        anc1 += sqrt(d)
        anc2 += anc1 ^ 3
        anc3 += G * mb
        anc4 += anc3 / anc2
        rb -= ra
    end
    y! += anc4 * rb
    ~@routine
    # rb is not recovered rigorously!
end
```

## G   Language Description

### G.1   Grammar

A minimum definition of NiLang's grammar is

| | |
|---|---|
| $s$ | a statement |
| $\bar{s}$ | multiple statements |
| $c$ | a constant |
| $x, y, z$ | symbols |
| $i, n$ | integers |
| $e$ | julia expression |
| $\sigma_P[x \mapsto v]$ | environemnt with $x$'s value equal to $v$ |
| $\sigma_P[x \mapsto nothing]$ | environemnt with $x$ undefined |
| $\sigma_P, e \Downarrow_e v$ | a Julia expression $e$ under environment $\sigma_P$ is interpreted as value $v$ |
| $\sigma_P, s \Downarrow_p \sigma'_P$ | the evaluation of a statement $s$ under environment $\sigma_P$ generates environment $\sigma'_P$ |
| $\sigma_P, s \Downarrow_p^{-1} \sigma'_P$ | the reverse evaluation of a statement $s$ under environment $\sigma_P$ generates environment $\sigma'_P$ |

$$
\begin{aligned}
\text{Statements} \quad s \quad ::= \quad &\sim s \mid e\,(\,d*\,) \mid x \leftarrow e \mid x \rightarrow e \\
&\mid \textbf{@routine } s\,;\,s*\,;\,\sim\textbf{@routine} \\
&\mid \textbf{if } (\,e\,,\,e\,)\,s*\,\textbf{else } s*\,\textbf{end} \\
&\mid \textbf{while } (\,e\,,\,e\,)\,s*\,\textbf{end} \\
&\mid \textbf{for } x = e : e : e\,s*\,\textbf{end} \\
&\mid \textbf{begin } s*\,\textbf{end} \\
\text{Data views} \quad d \quad ::= \quad &d\,.\,x \mid d\,[\,e\,] \mid d \triangleright e \\
&\mid c \mid x \\
\text{Reversible functions} \quad p \quad ::= \quad &\textbf{@i function } x\,(\,x*\,)\,\textbf{s* end}
\end{aligned}
$$

$\triangleright$ is the pipe operator in Julia. Here, $e$ is a reversible function and $d \triangleright e$ represents a **bijection** of $d$. Function arguments are data views, where a data view is a modifiable memory. It can be a variable, a field of a data view, an array element of a data view, or a bijection of a data view.

### G.2   Operational Semantics

The following operational semantics for the forward and backward evaluation shows how a statement is evaluated and reversed.

$$
\text{ANCILLA } \frac{\sigma_P, e \Downarrow_e v}{\sigma_P[x \mapsto nothing], x \rightarrow e \Downarrow_p \sigma_P[x \mapsto v]} \qquad \frac{\sigma_P, e \Downarrow_e v}{\sigma_P[x \mapsto v], x \leftarrow e \Downarrow_p \sigma_P[x \mapsto nothing]}
$$

$$
\text{ANCILLA}^{-1} \frac{\sigma_P, x \leftarrow e \Downarrow_p \sigma'_P}{\sigma_P, x \rightarrow e \Downarrow_p^{-1} \sigma'_P} \qquad \frac{\sigma_P, x \rightarrow e \Downarrow_p \sigma'_P}{\sigma_P, x \leftarrow e \Downarrow_p^{-1} \sigma'_P}
$$

$$
\text{BLOCK } \frac{\sigma_P, s_1 \Downarrow_p \sigma'_P \quad \sigma'_P, \textbf{begin } s_2 \cdots s_n \textbf{ end} \Downarrow_p \sigma''_P}{\sigma_P, \textbf{begin } s_1 \cdots s_n \textbf{ end} \Downarrow_p \sigma''_P} \qquad \frac{}{\sigma_P, \textbf{begin end} \Downarrow_p \sigma_P}
$$

$$
\text{BLOCK}^{-1} \frac{\sigma_P, s_n \Downarrow_p^{-1} \sigma'_P \quad \sigma'_P, \textbf{begin } s_1 \cdots s_{n-1} \textbf{ end} \Downarrow_p^{-1} \sigma''_P}{\sigma_P, \textbf{begin } s_1 \cdots s_n \textbf{ end} \Downarrow_p^{-1} \sigma''_P} \qquad \frac{}{\sigma_P, \textbf{begin end} \Downarrow_p^{-1} \sigma_P}
$$

$$
\text{FOR } \frac{\sigma_P, e_1 \Downarrow_e n_1 \quad \sigma_P, e_2 \Downarrow_e n_2 \quad \sigma_P, e_3 \Downarrow_e n_3 \quad (n_1 <= n_3) == (n_2 > 0) \quad \sigma_P[x \mapsto n_1], \bar{s} \Downarrow_p \sigma'_P \quad \sigma'_P, \textbf{for } x = e_1 + e_2 : e_2 : e_3\ \bar{s}\ \textbf{end} \Downarrow_p \sigma''_P}{\sigma_P, \textbf{for } x = e_1 : e_2 : e_3\ \bar{s}\ \textbf{end} \Downarrow_p \sigma''_P}
$$

$$\text{FOR-EXIT } \frac{\sigma_P, e_1 \Downarrow_e n_1 \quad \sigma_P, e_2 \Downarrow_e n_2 \quad \sigma_P, e_3 \Downarrow_e n_3 \quad (n_1 <= n_3) \mathrel{!=} (n_2 > 0)}{\sigma_P, \mathbf{for}\ x = e_1 : e_2 : e_3 \ \overline{s} \ \mathbf{end} \ \Downarrow_p \sigma_P}$$

$$\text{FOR}^{-1} \frac{\sigma_P, \mathbf{for}\ x = e_3 : -e_2 : e_1 \ \sim\!\mathbf{begin}\ \overline{s}\ \mathbf{end} \ \ \mathbf{end} \ \Downarrow_p \sigma'_P}{\sigma_P, \mathbf{for}\ x = e_1 : e_2 : e_3 \ \overline{s} \ \ \mathbf{end} \ \Downarrow_p^{-1} \sigma'_P}$$

$$\text{IF-T } \frac{\sigma_P, e_1 \Downarrow_e true \quad \sigma_P, s_1 \Downarrow_p \sigma'_P \quad \sigma'_P, e_2 \Downarrow_e true}{\sigma_P, \mathbf{if}\ (e_1, e_2)\ s_1\ \mathbf{else}\ s_2\ \mathbf{end} \ \Downarrow_p \sigma'_P}$$

$$\text{IF-F } \frac{\sigma_P, e_1 \Downarrow_e false \quad \sigma_P, s_2 \Downarrow_p \sigma'_P \quad \sigma'_P, e_2 \Downarrow_e false}{\sigma_P, \mathbf{if}\ (e_1, e_2)\ s_1\ \mathbf{else}\ s_2\ \mathbf{end} \ \Downarrow_p \sigma'_P}$$

$$\text{IF}^{-1} \frac{\sigma_P, \mathbf{if}\ (e_2, e_1)\ \sim\!\mathbf{begin}\ s_1\ \mathbf{end}\ \mathbf{else}\ \sim\!\mathbf{begin}\ s_2\ \mathbf{end}\ \mathbf{end} \ \Downarrow_p \sigma'_P}{\sigma_P, \mathbf{if}\ (e_1, e_2)\ s_1\ \mathbf{else}\ s_2\ \mathbf{end} \ \Downarrow_p^{-1} \sigma'_P}$$

$$\text{WHILE } \frac{\sigma_P, e_1 \Downarrow_e true \quad \sigma_P, e_2 \Downarrow_e false \quad \sigma_P, \overline{s} \Downarrow_p \sigma'_P \quad \sigma'_P, (e_1, e_2, \overline{s}) \Downarrow_{loop} \sigma''_P}{\sigma_P, \mathbf{while}\ (e_1, e_2)\ \overline{s}\ \mathbf{end} \ \Downarrow_p \sigma'_P}$$

$$\text{WHILE-REC } \frac{\sigma_P, e_2 \Downarrow_e true \quad \sigma_P, e_1 \Downarrow_e true \quad \sigma_P, \overline{s} \Downarrow_p \sigma'_P \quad \sigma'_P, (e_1, e_2, \overline{s}) \Downarrow_{loop} \sigma''_P}{\sigma_P, (e_1, e_2, \overline{s}) \Downarrow_{loop} \sigma''_P}$$

$$\text{WHILE-EXIT } \frac{\sigma_P, e_2 \Downarrow_e true \quad \sigma_P, e_1 \Downarrow_e false}{\sigma_P, (e_1, e_2, \overline{s}) \Downarrow_{loop} \sigma_P}$$

$$\text{WHILE}^{-1} \frac{\sigma_P, \mathbf{while}\ (e_2, e_1)\ \sim\!\mathbf{begin}\ \overline{s}\ \mathbf{end}\ \mathbf{end} \ \Downarrow_p \sigma'_P}{\sigma_P, \mathbf{while}\ (e_1, e_2)\ \overline{s}\ \mathbf{end} \ \Downarrow_p^{-1} \sigma'_P}$$

$$\text{UNCOMPUTE } \frac{\sigma_P, s \Downarrow_p^{-1} \sigma'_P}{\sigma_P, \sim\!s \ \Downarrow_p \sigma'_P}$$

$$\text{COMPUTE-COPY-UNCOMPUTE } \frac{\sigma_P, s_1 \Downarrow_p \sigma'_P \quad \sigma'_P, \mathbf{begin}\ \overline{s}\ \mathbf{end} \Downarrow_p \sigma''_P \quad \sigma''_P, s_1 \Downarrow_p^{-1} \sigma'''_P}{\sigma_P, @\mathbf{routine}\ s_1; \ \overline{s}; \ \sim\!@\mathbf{routine} \ \Downarrow_p \sigma'''_P}$$

$$\text{COMPUTE-COPY-UNCOMPUTE}^{-1} \frac{\sigma_P, @\mathbf{routine}\ s_1; \ \sim\!\mathbf{begin}\ \overline{s}\ \mathbf{end}\ \sim\!@\mathbf{routine} \ \Downarrow_p \sigma'_P}{\sigma_P, @\mathbf{routine}\ s_1; \ \overline{s}; \ \sim\!@\mathbf{routine} \Downarrow_p^{-1} \sigma'_P}$$

$$\text{CALL } \frac{\begin{array}{c}\sigma_P, d_i \Downarrow_{get} v_i \\ \varnothing[x_1 \mapsto v_1 \cdots x_n \mapsto v_n], (x_1 \cdots x_n) = x_f(x_1 \cdots x_n) \Downarrow_e \sigma_{P_0}[x_1 \mapsto v'_1 \cdots x_n \mapsto v'_n] \\ \sigma_{P_{i-1}}, v'_i, d_i \Downarrow_{set} \sigma_{P_i}\end{array}}{\sigma_P, x_f(d_1 \cdots d_n) \Downarrow_p \sigma_{P_n}}$$

$$\text{CALL}^{-1} \frac{\sigma_P, (\sim\!x_f)(d_1 \cdots d_n) \Downarrow_p \sigma'_P}{\sigma_P, x_f(d_1 \cdots d_n) \Downarrow_p^{-1} \sigma'_P}$$

$$\text{GET-VIEW } \frac{\sigma_P, d \Downarrow_e v}{\sigma_P, d \Downarrow_{get} v} \qquad\qquad \text{SET-VIEW-SYM } \frac{}{\sigma_P, v, x \Downarrow_{set} \sigma_P[x \mapsto v]}$$

SET-VIEW-ARRAY $\dfrac{z\ fresh \quad \sigma_P[z \mapsto v], setindex!(d,z,x) \Downarrow_e v' \quad \sigma_P, v', d \Downarrow_{set} \sigma'_P}{\sigma_P, v, d[x] \Downarrow_{set} \sigma'_P}$

SET-VIEW-FIELD $\dfrac{z\ fresh \quad \sigma_P[z \mapsto v], chfield(d,x,z) \Downarrow_e v' \quad \sigma_P, v', d \Downarrow_{set} \sigma'_P}{\sigma_P, v, d.x \Downarrow_{set} \sigma'_P}$

SET-VIEW-BIJECTOR $\dfrac{z\ fresh \quad \sigma_P[z \mapsto v], z \triangleright (\sim e) \Downarrow_e v' \quad \sigma_P, v', d \Downarrow_{set} \sigma'_P}{\sigma_P, v, d \triangleright e \Downarrow_{set} \sigma'_P}$

Here, $chfield(x,y,z)$ is a Julia function that returns an object similar to $x$, but with field $y$ modified to value $z$, $setindex!(x,z,y)$ is a Julia function that sets the $y$th element of an array $x$ to the value of $z$. We do not define primitive instructions like $d_1 \odot= x_f(d_2 \cdots d_n), \odot \in \{+,-,*,/,\veebar\}$ because these instructions are evaluated as a regular julia expression $prime(\odot, x_f)(d_1, d_2 \cdots d_n) = (d_1 \odot x_f(d_2 \cdots d_n), d_2 \cdots d_n)$ using the above CALL rule, where $prime(\odot, x_f)$ is a predefined Julia function. To reverse the call, the reverse julia function $\sim prime(\odot, x_f)$ should also be properly defined. On the other side, any non-primitive NiLang function defintion will generate two Julia functions $x_f$ and $\sim x_f$ so that it can be used in a recursive definition or other reversible functions. When calling a function, NiLang does not allow the input data views mappings to the same memory, i.e. shared read and write is not allowed. If the same variable is used for shared writing, the result might incorrect. If the same variable is used for both reading and writing, the program will become irreversible. For example, one can not use `x -= x` to erase a variable, while coding `x.y -= x.z` is safe. Even if a variables is used for shared reading, it can be dangerous in automatic differentiating. The share read of a variable induces a shared write of its gradient in the adjoint program. For example, `y += x * x` will not give the correct gradient, but its equivalent forms `z ← 0; z += x; y += x * z; z -= x; z → 0` and `y += x ^2` will.

### G.3 COMPILATION

The compilation of a reversible function to native Julia functions is consisted of three stages: *preprocessing*, *reversing* and *translation* as shown in Fig. 7.

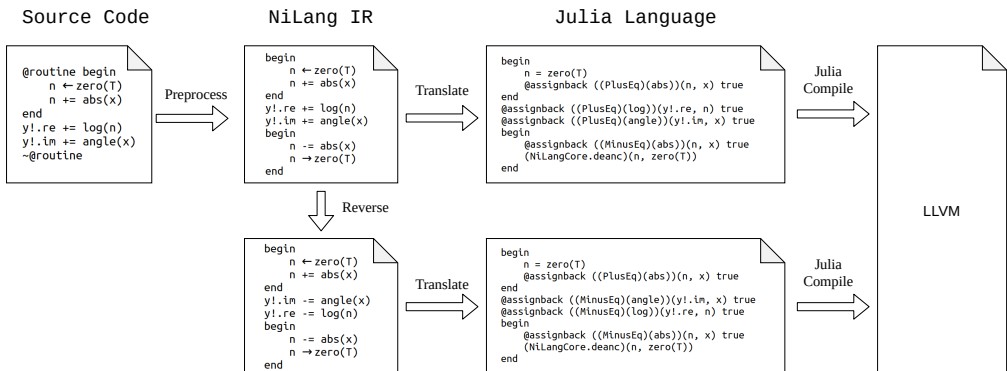

Figure 7: Compiling the body of the complex valued log function defined in Listing. 5.

In the *preprocessing* stage, the compiler pre-processes human inputs to the reversible NiLang IR. The preprocessor removes eye candies and expands shortcuts to symmetrize the code. In the left most code box in Fig. 7, one uses `@routine <stmt>` statement to record a statement, and `~@routine` to insert the corresponding inverse statement for uncomputing. This macro is expanded in the *preprocessing* stage. In the *reversing* stage, based on this symmetrized reversible IR, the compiler generates reversed statements. In the *translation* stage, the compiler translates this reversible IR as well as its inverse to native Julia code. It adds `@assignback` before each function call, inserts codes for reversibility check, and handle control flows. The `@assignback` macro assigns the outputs of a function to its input data views. As a final step, the compiler attaches a return statement that returns all updated input arguments. Now, the function is ready to execute on the host language.

