# OpenReview forum: "Differentiate Everything with a Reversible Domain-Specific Language"
_ICLR.cc/2021/Conference — Reject_

### Official Review · AnonReviewer2 · 2020-10-24
**A promising manuscript on a new approach that combines reversible programming and automatic differentiation; Call for improvements**

**Rating:** 6
**Confidence:** 2

**Review:**

General comments:
- This paper presents a new approach to automatic differentiation (AD), namely the use of reversible programming to achieve memory-efficient function inverse and adjoint. The authors have done a good job reviewing the background and laying out the motivation for the new apporach. The implementation is based on adding an embedded DSL to Julia called NiLang. Through reversible programming, NiLang gets rid of the need for checkpointing and hence is amenable to CUDA execution. NiLang is benchmarked against native Julia, ForwardDiff and Tapenade. The performance of NiLang is slightly worse than other approachs in the GMM benchmark. But in the bundle adjustment benchmark, NiLang outperforms ForwardDiff and Tapenade, especially with CUDA acceleration.

Detailed comments:
- Section 2.3
  - I feel the explanation of the pre and post conditions can be made clearer. While most readers are familiar with the idea of pre condition in if and while control flows, some may not be familiar with what post condition is. The example in Listing 4 isn't presented in a way that's most clear and helpful to resolve this confusion either. Please explain post condition more clearly, perhaps be adding annotations to Listing 4.
  - Also, in Figure 1(a), what does "pre == post" mean? On a related note, in Listing 4, if the post condition of the if statement is a placeholder "~", how does it work when it's treated as the pre condition during reversal? Explain that it's simply treated as an always-true condition.
  - Please add a sentences to explain how NiLang handles the errors (as shown in red in Figure 1) that occur during the control flow and their reversals.
- Section 3.1
  - Please explain what adjoint is briefly before mentioning it first time, possibly by citing the Tapenade paper (Hascoet & Pascual 2013).
- Section 4.1
  - Since the authors emphasize the memory usage advantage of NiLang, why not run memory benchmarking, quantify the results and show them in tables or figures here?
  - Table 1: Why are NiLang GPU results not included in this table?
  - Tables 1 and 2: The timing numbers in these tables to not span many orders of manitude. So using regular decimal points (e.g., 0.009844, 0.0351) might be more visually clear and easier to parse than the engineering notation currently used. Also, consider making plots instead of tables for these numbers, because plots will be much more intuitive and facilitate comparisons between the different implementations. The plots can be in logarithmic scale.
- This manuscript currently lacks a discussion section. Given AD is widely used in machine learning and neural networks (especially within the context of this conference), many readers will be interested in whether NiLang is suitable for training neural networks.

More detailed comments / writing suggestions:
- p 2. "Besides the above “nice” features, it also has some “bad” features to meet the practical needs" The meaning of the second part of this sentence is unclear. Please rewrite it.
- p 8, Section 4.1. Do not repeatedly define the ancronym "BA". It's already defined above in the same page. Also, since you don't really use the acronym anyway, there's no need to define it.

---

> ### Author Response · Authors · 2020-11-15
> **Memory analysis et. al.**
>
> Thanks for your careful reading. I revised the paper according to your comments and now the paper looks much better!
>
> Major changes:
>
> > I feel the explanation of the pre and post conditions can be made clearer. While most readers are familiar with the idea of pre condition in if and while control flows, some may not be familiar with what post condition is. The example in Listing 4 isn't presented in a way that's most clear and helpful to resolve this confusion either. Please explain post condition more clearly, perhaps be adding annotations to Listing 4.
> Also, in Figure 1(a), what does "pre == post" mean? On a related note, in Listing 4, if the post condition of the if statement is a placeholder "~", how does it work when it's treated as the pre condition during reversal? Explain that it's simply treated as an always-true condition.
>
> 1. The precondition and postconsiditon are not explained well, so I added 4 pieces of speudocode (listings 4-8) to explain how it is translated to the host language. I feel they are more explicit than the diagram.
>
> > Since the authors emphasize the memory usage advantage of NiLang, why not run memory benchmarking, quantify the results and show them in tables or figures here?
>
> We added the memory analysis to the GMM benchmark (Fig. 4), where the irreversible program shows negligible overhead in memory.
> Since peak memory analysis is hard in general, we do not do the same analysis for Tapenade.
>
> > Table 1: Why are NiLang GPU results not included in this table? Tables 1 and 2: The timing numbers in these tables to not span many orders of manitude. So using regular decimal points (e.g., 0.009844, 0.0351) might be more visually clear and easier to parse than the engineering notation currently used. Also, consider making plots instead of tables for these numbers, because plots will be much more intuitive and facilitate comparisons between the different implementations. The plots can be in logarithmic scale.
>
> We changed the talbes to plots. However, we can now show the GMM benchmark on GPU because the code is not writen in a GPU compatible style. Writting GPU kernels for GMM might be possible (https://core.ac.uk/download/pdf/206684904.pdf), but requires a lot extra efforts. On the otherside, loading BA code to GPU requires no more than 10 lines of extra code.
>
> > This manuscript currently lacks a discussion section. Given AD is widely used in machine learning and neural networks (especially within the context of this conference), many readers will be interested in whether NiLang is suitable for training neural networks.
>
> Discussion section added.
>
> Also, please notice the newly added section 2 (see the reply to reviewer 1). It discusses the time-space tradeoff in reversible programming and checkpointing from a higher level, which might be helpful to users to understand the gist of reversible programming AD.
>
> Thanks again and looking forward to your feedback!

---

### Official Review · AnonReviewer4 · 2020-10-25
**This paper presents a Julia based DSL to do automatic differentiation using reversible computing**

**Rating:** 4
**Confidence:** 4

**Review:**

The paper adapts reversible computing techniques to compute gradients. The techniques presented are not new though the Julia based DSL is new. The results presented are for differentiating through a GMM. It is not clear if the technique scale to a modern day neural network models and how they will integrate into current frameworks like JAX, PyTorch or TensorFlow.

The paper may be more relevant to a Programming Language focused venue or maybe even JuliaCon

---

> ### Author Response · Authors · 2020-11-10
> **NiLang is designed for differential programming**
>
> > The techniques presented are not new
>
> Due to the limited knowledge of authors, we don't know any other works in the field of automatic differentiation use reverse computing. Works I know are all based on checkpointing, which is [different to reverse computing](https://nextjournal.com/giggle/reverse-checkpointing). If you know any of such work, please let us know, because we need to revise our writing a lot then.
>
> > It is not clear if the technique scale to a modern day neural network models and how they will integrate into current frameworks like JAX, PyTorch or TensorFlow.
>
> Since our work is based on the Julia language, it can not be directly used in python. But it can be used in Julia machine learning frameworks like Zygote to accelerate the backward rule of norm2 function by 10^3 ([link](https://giggleliu.github.io/NiLang.jl/dev/examples/port_zygote/#How-to-port-NiLang-to-Zygote )). Due to the page limit, we can not present this part in the main text. But it sounds like a good idea to add an appendix to explain this point better.
>
> > The paper may be more relevant to a Programming Language focused venue or maybe even JuliaCon
>
> The content covers a lot about language level design, but the aim is for differential programming. I think this paper fits quite well with the ambitious goal in the deep learning field to differentiate everything
>
> https://techburst.io/deep-learning-est-mort-vive-differentiable-programming-5060d3c55074
>
>  [swift language](https://tryolabs.com/blog/2020/04/02/swift-googles-bet-on-differentiable-programming/) and [Taichi language](https://dl.acm.org/doi/10.1145/3355089.3356506) are predecessors in the machine learning field towards this direction.
>
> But we agree the paper should not focus too much on the implementation detail and language specific stuff. We revise the paper by addinga new section (Sec. 2) and discuss AD from a higher level rather than focusing the implementation details.

---

### Official Review · AnonReviewer3 · 2020-10-30

**Rating:** 5
**Confidence:** 3

**Review:**

# Changes after rebuttal
Thanks to the authors for their answers to the questions and their revisions to improve the manuscript. It is useful to have further descriptions of reversible computing for an audience that may be unfamiliar with the topic. I would encourage the authors to make further revisions to more concisely show the scientific value of the work while leaving some of the details to tutorials or other documents.  Other venues more focused on scientific computing, programming languages, or Julia may also be more suitable. If the language also attracts more users and applications built on top of it, then the case for publication will also be stronger (consider that the PyTorch paper was presented at NeurIPS 2019 even though the first release was in 2016).

---
# Summary
The paper presents an embedded domain-specific language in Julia which enables reversible computation and automatic differentiation. In order to compute gradients, we need access to all the intermediate results in a program; typically, it is necessary to store (checkpoint) these intermediate results separately or recompute them from the inputs. With reversible computing, these are unnecessary as we can compute backwards from the output to reach any intermediate result. The paper shows some empirical performance benchmarks on a bundle adjustment program.

# Strengths
- The proposed system shows strong performance compared to the presented alternatives, especially when using GPU kernels.
- There are many examples presented which shows that the language is terse.
- The system is practicality and usable for existing scientific computing applications.

# Weaknesses
- As a significant portion of the paper is dedicated to examples of the language in use, there is not much room to discuss other aspects of the system.
- The novel aspects of the work are not made very clear in the paper. It is hard to compare the benefits and drawbacks of the work compared to the alternatives, other than in the runtime performance.
- ICLR is about machine learning, but there is no evaluation of machine learning workloads. For example, I think the ICLR audience would be quite interested in how such a system might enable very deep neural networks.

# Comments
- For people not very familiar with reversible computing (like me), it is quite useful to have detailed examples and explanations about the fundamentals. However, it seemed to me that too much of the main body is spent on the fundamentals and the tutorial aspects, and more of it could be moved into an appendix.
- It would be better to have more comparisons and discussions of which aspects of the system are novel. There is a discussion of some related work on page 2, but it is not very systematic.
- As the paper states at the end of page 2, it is not possible to rigorously reverse operations on floating point numbers. However, there is no further discussion of the implications of this. It would be good to have some further reassurance that the errors are not important, or to have a discussion of what applications are acceptable and sufficiently tolerant of the errors.
- Given the lack of checkpointing required for automatic differentiation, it would seem to me that the language can enable significantly lower memory usage than non-reversible languages. So I was surprised that there are no benchmarks discussing memory usage, or showing that the method can operate with larger datasets/parameters on a fixed memory budget.
- Anonymous submissions should not include acknowledgements and should not include links to non-anonymous GitHub repositories.
- Please use \citet and \citep with natbib so that citations are formatted properly in the text. Citations at the end of sentences, or otherwise not used as a noun in the sentence, should look like (Author, 2020) rather than Author (2020).

---

> ### Author Response · Authors · 2020-11-17
> **Response**
>
> Thank you for your careful review. I agree that I focused too much on the language features, now we have moved most of them to the appendix and focus more on the high-level descriptions.
>
> > The novel aspects of the work are not made very clear in the paper. It is hard to compare the benefits and drawbacks of the work compared to the alternatives, other than in the runtime performance.
>
> The novelty of this paper is, it proposes a different scheme, reverse computing, as an alternative to the checkpointing that is used in previous AD and machine learning packages. I added Sec.2 to describe this point better.
>
> > ICLR is about machine learning, but there is no evaluation of machine learning workloads. For example, I think the ICLR audience would be quite interested in how such a system might enable very deep neural networks.
>
> Both examples in the benchmarks, Gaussian mixture models and Bundle adjustment, are machine learning applications. But we agree that many people will be interested if it focuses more on deep neural networks. We added some discussion at the end of the main text, where we mentioned an arbitrary deep NICE network can be differentiated in constant memory with NiLang. NiLang is most useful in deep learning as an adjoint rule generator.
>
> > As the paper states at the end of page 2, it is not possible to rigorously reverse operations on floating-point numbers. However, there is no further discussion of the implications of this. It would be good to have some further reassurance that the errors are not important, or to have a discussion of what applications are acceptable and sufficiently tolerant of the errors.
>
> We added this part as an appendix F. Where we use the leapfrog integrator to experiment with the rounding error. NiLang can backpropagate this program in constant memory, hence the number of depths can easily scale up to >2^25 without suffering from the rounding errors.
>
> > Given the lack of checkpointing required for automatic differentiation, it would seem to me that the language can enable significantly lower memory usage than non-reversible languages. So I was surprised that there are no benchmarks discussing memory usage, or showing that the method can operate with larger datasets/parameters on a fixed memory budget.
>
> It is a great suggestion. We added a new figure for the GMM benchmark in the main text (Fig. 4) that compares the peak memory with its irreversible counterpart. Notice measuring the peak memory is not easy, and the difference is so small that the statistic methods can not show the difference. We counted the allocations manually because the benchmarked code is not complicated. If you feel the counting is unreliable, we can provide the source code, where every allocation is explicitly written.
>
> > Anonymous submissions should not include acknowledgments and should not include links to non-anonymous GitHub repositories.
>
> We feel sorry for this. Now I have removed the links and acknowledgments.
>
> > Please use \citet and \citep with natbib so that citations are formatted properly in the text. Citations at the end of sentences, or otherwise not used as a noun in the sentence, should look like (Author, 2020) rather than Author (2020).
>
> We changed them to \citep. This is very helpful. Thanks for telling us this trick!

---

### Official Review · AnonReviewer1 · 2020-11-02
**Interesting contribution but not clearly presented**

**Rating:** 6
**Confidence:** 4

**Review:**

This paper draws connections between reversible programming and reverse mode automatic differentiation and introduces a reversible DSL in Julia that can be used to calculate first and second order gradients.

I reviewed a previous version of this paper for NeurIPS.

I really like the idea of reversible programming and I think that a clear introduction of reversible programming and its use in automatic differentiation could be of interest to the machine learning community. However, I feel that this paper fails to clearly explain the use of reversible programming and its trade-offs compared to checkpointing and other existing approaches.

As a paper, the first section is great, but then the authors leave me with many questions: How do checkpointing and reversible programming differ in memory usage? Given that the multiplier from listing 1 has 3 outputs, doesn't that mean that a program consisting of n chained multiplications still requires storing n * 2 + 1 outputs, similarly to regular AD? And doesn't binomial checkpointing allow for logarithmic memory usage in exchange for a logarithmic increase in runtime (rather than polynomial)?

Rather than answering these questions, the paper jumps eagerly into Julia code snippets, metaprogramming, and CUDA kernels, which I don't feel actually serve to elucidate the message that reversible programming is of interest to the machine learning community.

Although I feel that this version of the paper is an improvement over the version I reviewed for NeurIPS, I feel that it still fails to clearly introduce reversible programming and shed light on the subtle trade-offs between reversible programming, checkpointing, and regular AD. I encourage the authors to rewrite the paper with less of a focus on the implementation details of their framework, and a stronger focus on the memory and runtime trade-offs provided by all of these methods from a more high-level, theoretical perspective.

Pros

* Very relevant and interesting topic
* Well-written introduction
* Good code

Cons

* Fails to introduce the topic appropriately for an ML audience
* Does not clearly compare to advanced checkpointing methods
* Not well written; too many details about the software implementation that do not contribute to an understanding of the high-level technique

---

> ### Author Response · Authors · 2020-11-13
> **Rewrite part of the paper to focus on higher level discussions**
>
> Thank you for reviewing my paper two times, we appreciate your every single comment.
>
> > As a paper, the first section is great, but then the authors leave me with many questions: How do checkpointing and reversible programming differ in memory usage? Given that the multiplier from listing 1 has 3 outputs, doesn't that mean that a program consisting of n chained multiplications still requires storing n * 2 + 1 outputs, similarly to regular AD? And doesn't binomial checkpointing allow for logarithmic memory usage in exchange for a logarithmic increase in runtime (rather than polynomial)?
>
> You are right.  We added a section named "2. Reverse computing as an Alternative of Checkpointing" in the main text to compare checkpointing and reverse computing. The treeverse algorithm for optimal checkpointing and the Bennett's time-space trade-off algorithm in reverse computing are two different things, reviewing two different schemes and comparing their complexity in different senarios deepens our discussion.
>
> Implementing chained multiplications with constant memory-space overheads is one of the examples that we use demonstrate the advantage of reverse computing. Now it is in Appendix B.3. The reason why reverse computing shows advantage is because we can use logarithmic numbers that reversible under `*=`. for chained multiplication. Same trick can be used in computing taylor expansions.
> If we do not utilize logarithmic numbers, the binomial checkpointing is better than reverse computing, because reverse computing suffers from a polynomial overhead to achieve logarithmic space overhead.
>
> > I encourage the authors to rewrite the paper with less of a focus on the implementation details of their framework, and a stronger focus on the memory and runtime trade-offs provided by all of these methods from a more high-level, theoretical perspective.
>
> Except the above added section. We also moved the implementation details of automatic differentiation and many other code snipetts to the appendix, and explain the implementation of AD from a higher level.
>
> We are lucky to have you reviewing our paper, your comments improved our paper significantly. Looking forward to your feedbacks.

---

### Official Review · AnonReviewer5 · 2020-11-23
**Evaluation and presentation needs to be improved**

**Rating:** 5
**Confidence:** 3

**Review:**

I apologize for the late review since I did not realize I was assigned an emergency review for this paper.

## Summary

The paper proposes a new DSL language embedded in Julia that can represent reversible programs. They provide mechanisms to automatically reverse a program given its forward definition in their DSL. Their language would allow reverse mode automatic differentiation to be implemented in lieu of the standard checkpointing schemes.

## Strengths

* A DSL to express reversible programs and implementation in a well-known language
* Simple description

## Weaknesses

* Very limited evaluation on two benchmarks.
* Presentation issues that distract from the main point of the paper.
* Vague language description and limitations not mentioned.


## General discussion and Questions for the authors

I liked the high-level idea of this paper, however the presentation and writing need to be drastically improved in order to be accepted for publication. Also, the current results section is limited to two examples and I recommend adding more case studies to validate the usefulness of their DSL.

At a high-level the authors can do the following to improve their presentation.

* Some of the language implementation details distract the readers from understanding the design aspects. For example. “One can input “←” and “→” by typing “\leftarrow[TAB KEY]” and “\rightarrow[TAB KEY]” respectively in a Julia editor or REPL”. The authors can omit the implementation details or move it to appendices and focus more on the design decisions they took and the reasoning for those decisions.
* The authors can present their DSL language more formally. For example, they can give operational semantics for their forward program and reversed program using deductive rules.
* What’s supported by the language and what’s not is not clearly mentioned. Specifically, the authors should mention limitations and scope of the language.

Evaluation needs to be improved

* It is not conclusive from the results presented that the DSL (or reverse mode in general) produces better code than a checkpointing strategy. It helps in case of bundle adjustment, however has an overhead in the GMM implementation.
* Coverage - To showcase the expressivity of the DSL, the authors should implement a known NN architecture (e.g., even a small resnet) and show how training and inference speeds vary.

---

> ### Author Response · Authors · 2020-11-23
> **The reply to Reviewer #5**
>
>
> > I liked the high-level idea of this paper, however the presentation and writing need to be drastically improved in order to be accepted for publication. Also, the current results section is limited to two examples and I recommend adding more case studies to validate the usefulness of their DSL.
>
> Thanks for taking time to review our paper. We will try our best to present it better.
>
> * Some of the language implementation details distract the readers from understanding the design aspects. For example. “One can input “←” and “→” by typing “\leftarrow[TAB KEY]” and “\rightarrow[TAB KEY]” respectively in a Julia editor or REPL”. The authors can omit the implementation details or move it to appendices and focus more on the design decisions they took and the reasoning for those decisions.
>
> For people interested in the theory part, we suggest only reading the introduction and section two. The rest is about engineering. Most reasoning and decisions are quite straight-forward.
>
> Our work includes some language details, including some tutorial purposed. Because, we structured the code snippets in the main text in such a way that, if a user pastes the code into a Julia REPL, it executes
> https://asciinema.org/a/A3piE4d664l4BEqTqobpat8FD .
> We believe, allowing a user to feel how the reversible instructions, control flows and memory management executes is more valuable than describing it vividly. Then it is necessary for user to know how to input left and right arrows, and we wish it can be a part of the main text because we presume readers are not familiar with Julia.
>
> > The authors can present their DSL language more formally. For example, they can give operational semantics for their forward program and reversed program using deductive rules.
>
> You are right. We added an appendix G to describe the compiling process, grammar and the operational semantics.
>
> > * What’s supported by the language and what’s not is not clearly mentioned. Specifically, the authors should mention limitations and scope of the language.
> *  It is not conclusive from the results presented that the DSL (or reverse mode in general) produces better code than a checkpointing strategy. It helps in case of bundle adjustment, however has an overhead in the GMM implementation.
> *  Coverage - To showcase the expressivity of the DSL, the authors should implement a known NN architecture (e.g., even a small resnet) and show how training and inference speeds vary.
>
> NiLang is not a replacement of any traditional machine learning packages, but a compliment. This is why we do not use resnet as an example. The reason for not doing this is, NiLang's linear algebra functions are very slow comparing with modern BLAS, and people probably do not want to repeat the effort to rewrite BLAS reversibly. In the benchmark of GMM, as we have mentioned in the main text, the computing time is dominated by the BLAS functions and NiLang does not optimize the memory layout for BLAS functions at the time of writing.
>
> Except BLAS functions, NiLang is very fast and very memory efficient in differentiating differential equations, quantum simulations et. al. A paper citing NiLang differentiated a 28x28 spinglass solver by re-writing quantum simulator with NiLang: https://arxiv.org/abs/2008.06888 . We do not think any other AD framework can do this.
> NiLang can be used in deep learning for generating backward rules for some non-standard functions, see Appendix E for an example of using NiLang to accelerate the backward rule of `norm` for Zygote - a popular machine learning package in Julia. We also benchmarked the backward rules for sparse matrix operations in appendix D.1, which could be useful building blocks in some neural networks.
>
> To avoid confusion, we added some statements in the discussion section to discourage people who wants to use NiLang to handle the computational graph in deep learning. We also added some other challenges for NiLang, where we think it can be improved.

---

### Author Response · Authors · 2020-11-25
**List of changes**

1. new section: Sec. 2 Reverse computing as an Alternative to Checkpointing - review and compare optimal checkpointing and optimal reverse computing. They turn out to be quite different. Reverse computing does no show advantage in the worst case complexity, but we argue it has advantage in practical using cases.
2. new section: Sec. 6: Discussion - summarize NiLang's pro and cons.
2. new appendix G: Language Description - describe the grammar, operational semantics and the compiling stages of NiLang.
3. new appendix F: A benchmark of round-off errors in Leapfrog - a new example of integrating the sympletic solar system, to benchmark the round-off errors introduced by different ways of writting a reversbile program. We show the NiLang does not nessesarily introduce additional rounding errors, the way of writing matters.
4. new appendix E: Porting NiLang to Zygote - showing NiLang can be useful as a primitive generator for Zygote.
4. move the implementation of AD in NiLang to the appendix, and only keep high level descriptions in the original section.
5. new figure 4, memory analysis.
6. change original tables to figure 3 for better visualization.

---

### Decision · Program_Chairs · 2021-01-07
**Final Decision**

**Decision:**

Reject

**Comment:**

After reading the paper, reviews and authors’ feedback. The meta-reviewer agrees with the reviewers that the paper touches an interesting topic (reversible computing) but could be improved in the area of presentation and evaluation. Therefore this paper is rejected.

Thank you for submitting the paper to ICLR.